# Simulation and Training of Needle Puncture Procedure with a Patient-Specific 3D Printed Gluteal Artery Model

**DOI:** 10.3390/jcm9030686

**Published:** 2020-03-04

**Authors:** Paweł Rynio, Aleksander Falkowski, Jan Witowski, Arkadiusz Kazimierczak, Łukasz Wójcik, Piotr Gutowski

**Affiliations:** 1Department of Vascular Surgery, Pomeranian Medical University in Szczecin, Powstańców Wielkopolskich 72, 70-111 Szczecin, Poland; Biker2000@wp.pl (A.K.); Piotr_gutowski@poczta.onet.pl (P.G.); 2Department of Radiology, Pomeranian Medical University in Szczecin, Powstańców Wielkopolskich 72, 70-111 Szczecin, Poland; bakhis@hot.pl (A.F.); woj.luka@gmail.com (Ł.W.); 32nd Department of General Surgery, Jagiellonian University Medical College, Kopernika 21, 31-501 Kraków, Poland; jan.witowski@alumni.uj.edu.pl

**Keywords:** 3D printing, simulation, endovascular procedure, gluteal artery, endoleak, needle puncture, vascular access

## Abstract

The puncture of the gluteal artery (GA) is a rare and difficult procedure. Less experienced clinicians do not always have the opportunity to practice and prepare for it, which creates a need for novel training tools. We aimed to investigate the feasibility of developing a 3D-printed, patient-specific phantom of the GA and its surrounding tissues to determine the extent to which the model can be used as an aid in needle puncture planning, simulation, and training. Computed tomography angiography scans of a patient with an endoleak to an internal iliac artery aneurysm with no intravascular antegrade access were processed. The arterial system, including the superior GA with its division branches, and pelvic area bones were 3D printed. The 3D model was embedded in the buttocks-shaped, patient-specific mold and cast. The manufactured, life-sized phantom was used to simulate the GA puncture procedure and was validated by 13 endovascular specialists. The printed GA was visible in the fluoroscopy, allowing for a needle puncture procedure simulation. The contrast medium was administered, simulating a digital subtraction angiography. Participating doctors suggested that the model could make a significant impact on preprocedural planning and resident training programs. Although the results are promising, we recommend that further studies be used to adjust the design and assess its clinical value.

## 1. Introduction

The gluteal artery (GA) is considered a viable vascular access point in situations when more favorable approaches cannot be used. The GA is small in diameter and located deep beneath subcutaneous tissue and gluteal muscle. Percutaneous access has been proven to be a feasible [1], although unfavorable method, as difficult anatomical features and few case-based examples make the procedure difficult.

Due to its proximity to the internal iliac artery (IIA), the GA plays an important role in IIA aneurysm (IIAA) pathophysiology. It is usually the source of a retrograde perfusion towards the IIAA or the source of type II endoleaks after endovascular aortoiliac aneurysm repair (EVAR) [2]. The first may occur in the case of embolization of the internal iliac artery (IIA) origin by a stent-graft implantation into the common and external iliac artery, thus covering the IIA origin. A non-embolized IIAA distal end may lead to retrograde sac pressurization and its subsequent growth. Although it is considered inappropriate now, many patients have been treated in this way in the past. A similar pathogenesis of aneurysm growth can be observed after an open repair of aortoiliac pathology when the IIA origin is ligated [1]. Endovascular aortoiliac aneurysm repair usually requires landing distally on the external iliac artery, leaving the IIA as a potential source of a type II endoleak. Mechanical blockage of the antegrade route requires another approach. In the event of such complications, the GA may be the only vascular access point that allows endoleak embolization. Transgluteal access enables full embolization of the IIAA feeding arteries.

Due to the difficulties related to the procedure, different adjuvant techniques have been proposed. Kabutey et al. used femoral access for angiographic road mapping of the GA and then carried out a fluoroscopic-guided puncture [1]. Another method of assessing the location of the GA in surrounding tissues is through the use of Doppler ultrasound [3]. This method, however, can be limited, especially in obese patients. Additional techniques to assist the needle puncture include the direct exposition of the GA. Magishi et al. performed a 10-cm incision in the buttock, split the gluteus maximus muscle, and inserted a 5-F introducer sheath into the inferior GA [4]. Attempts have been made to omit transgluteal access and directly puncture the IIAA sac instead. The IIAA puncture was carried out with the assistance of cone-beam computed tomography (CT) guidance [5]. In order to reduce the risk associated with a needle passing through the pelvis and thus the risk of damaging the vital structures and surrounding bowel, Gemmete et al. reported two cases of direct IIAA puncture using percutaneous transosseous approach [6]. The authors pointed out possible complications associated with the transosseous technique, including osteomyelitis, retroperitoneal hemorrhage, and pelvic bone fracture.

In addition to the difficulties related to gaining GA access, research on the topic is limited, due to the infrequent need for this intervention. This can result in resident doctors not having the opportunity to learn and get necessary experience. Thus, it is highly desirable to develop an appropriate tool that would facilitate training and personalize preoperative planning. In order to achieve this, we designed a simulator that would faithfully reflect patient anatomy and allow for training in a hybrid room. This study aimed to investigate the methodology of manufacturing a low-cost 3D replica of the GA and surrounding tissues based on the patient’s computed tomography data. We also attempted to determine the utility of such a model as an aid in pre-procedural planning and medical training, using questionnaires for endovascular specialists involved in the process.

## 2. Materials and Methods

### 2.1. Fabrication of the 3D Gluteal Artery Model

Retrospectively acquired CT angiography (CTA) images of a patient with an IIAA were used in this study (Figure 1A,B). The patient had undergone the EVAR procedure with a stent-graft landing distally on the external iliac artery, thus with no available antegrade access to the IIA. Continuous retrograde sac perfusion had caused aneurysm growth. The CT scans were performed at 0.75 mm slice thickness using a SOMATOM Definition AS scanner (Siemens Healthcare GmbH, Erlangen, Germany).

The arterial system (including aortic bifurcation, iliac arteries, IIAAs, and superior gluteal arteries with their division branches), pelvic bone, and skin of the pelvis and buttocks were segmented and exported as 3D mesh STL files with the use of Osirix MD 9.0 software (Pixmeo SARL, Bernex, Switzerland). Subsequently, mesh post-processing was executed in Blender software (Blender Foundation, Amsterdam, The Netherlands). The arterial model was hollowed, and the 1-mm-thick wall was added. The skin model formed the outside layer of the phantom, surrounding the pelvis. Its wall thickness was set to 1.5 mm. Next, the structures of the arterial system and bones were subtracted from the bottom and the top walls of the phantom, forming imprints of their shape. This stage was carried out to assemble 3D-printed structures similar to a jigsaw puzzle. Likewise, openings for the GA were made in the pelvic bone model.

The shell model was cut into several pieces to fit the 3D printer (Raise3D Pro 2, Shanghai, China) building volume (30 × 30 × 60 cm). The GA and IIAA endoleak models were 3D printed as one object (Figure 2A); the rubber-like filament was used in order to mimic arterial wall softness and flexibility. It had a Shore hardness of 95A (Filaflex Red, Recreus Industries S.L., Alicante, Spain). The arterial system and the bones were printed together during one print job using colored polylactic acid (PLA) filaments: (colorFabb, Bremweg, The Netherlands) (Figure 2B). Parts of the outside shell were manufactured with white PLA filament (Figure 2C,D). All models were 3D printed with a layer thickness of 0.2 mm. Printing temperatures were 230 °C for the flexible filament and 210 °C for the PLA. The printer bed was heated to 60 °C. To ensure the extrusion of flexible filament, we used a reduced printing speed of 30 mm/s in comparison to the 50 mm/s PLA printing speed. A smaller retraction distance of 0.5 mm also helped to print the flexible parts.

These 3D-printed parts were then assembled and glued together. Two inlet and outlet ducts were glued to the GA and IIAA model to allow contrast medium administration (Figure 2E,F). The ducts were routed outside the pelvic shell model. The model of the arteries and bones was inserted into the shell model, fitting inside the corresponding imprints. This step ensured that anatomical relations were maintained. The shell model served as a mold for casting, as suggested previously on hepatic 3D models by Witowski et al. [7]. In the upper wall of the shell layer, a hole was made for pouring transparent rubber silicone (Polastosil^®^ M-2000, Silikony Polskie Sp. zo.o., Nowa Sarzyna, Poland). The shell walls were reinforced with adhesive tape to prevent silicone leakage. Curing took 72 h at room temperature. The silicone, which had a Shore hardness of 30A, replicated subcutaneous tissue and muscles, allowing the anatomy of bones and arteries to be visually examined and their spatial relationships assessed in preparation for the GA puncture procedure. The last step was to remove the outer plastic shell, leaving the bones and arteries embedded in the silicone. The model was ready for tests under fluoroscopy. All stages were performed by a physician experienced in the evaluation of vascular CT scans and the creation of medical 3D models.

### 2.2. Model Validation

Thirteen endovascular interventionists, including interventional radiologists and vascular surgeons, tested the model; two of them were residents. The least experienced had 4 years of experience in the field. Each of the remaining specialists had 10 or more years of experience. They were asked to access the GA under conditions equivalent to a real endovascular procedure, working on an Artis zee angiography system (Siemens Healthcare GmbH, Erlangen, Germany). The choices of endovascular equipment and the procedure time were not subject to any limits. After the simulation, physicians completed a survey using the Likert scale (Figure 3).

## 3. Results

### 3.1. Three-Dimensional GA Phantom Fabrication

The final model was a faithful replica of the patient’s anatomy at a 1:1 scale (Figure 4A–E). The model consisted of the right GA connected with IIAA endoleak (soft parts), the common and external iliac arteries, the left IIA, the left GA and pelvic bones (rigid parts). Other soft tissues were recreated with soft transparent silicone. Segmentation and modelling took about 5 h. The total printing time was 131 h and the silicone curing took three days. It took no more than 1 h to bind each piece of the model together. The total manufacturing time was nine days. The total cost of the model was $430 (the filament cost about $80 and nearly 15 L of silicone for $350 were used).

### 3.2. Clinical Tests

The 3D-printed model was used during a mock GA puncture. Endovascular specialists performed the procedure in a hybrid room environment (Figure 5A,B). Physicians were given the option of studying the model before the procedure. The contrast could be administered through the inlet duct, which fed into the aneurysm of the IIA and the GA (Figure 5C). This step mimicked the in vivo administration of contrast with the guidance of digital subtraction angiography or 3D road mapping. Closing the outlet duct stopped the flow of contrast inside the model. The angiography angle manipulation also enabled physicians to choose the optimal vantage points, which may very well correspond to those used in the real procedure.

Moreover, it was possible to perform 3D rotational angiography and 3D road mapping. The physicians experimented with different places on the buttock surface to start the puncture. The operators punctured various parts of the GA, looking for the most convenient one. A transgluteal approach allowed for the contrast administration as well; however, while outside the vessel, the extravasation of the contrast could be seen on the screen, improving the realism of the simulation. Physicians expressed satisfaction with the model for the purposes of GA puncture training and simulation. The results of the survey are presented in Figure 3.

## 4. Discussion

Three-dimensional printing has been used previously for the guidance, navigation and planning of endovascular procedures [8]. The advancements that followed were focused on the design and utilization of 3D models as functional replicas. Itagaki et al. used a splenic artery model to simulate the endoluminal embolization procedure [9]. A set of catheters and guidewires were tested, and the most appropriate were selected for the real procedure. A similar simulation was carried out on the aorta to test the possibility of crossing unfavorable anatomy [10]. Based on the simulation, the operative plan was altered. Such simulations potentially lead to a reduction in the operation time, amount of endovascular equipment used, and thus in the cost of the procedure. Sterilized aortic replicas were used as a template to perform stent-graft modifications [11]. The presented GA model contributes to the existing body of research on functional replicas. It allows physicians to preview patient anatomy, simulate the procedure, identify possible complications, increase operator confidence, and choose appropriate equipment. This approach is in line with the trend of personalized medicine, i.e., tailored to the specific patient.

Physicians are expected to obtain consistent, positive treatment results. However, young doctors must also have opportunities to practice. Studies have demonstrated that resident involvement in treatment is independently associated with increased risk for major morbidity [12]. In light of negative results, there has been a push for new teaching methods. The introduction of training on simulators resulted in a paradigm shift from “see one, do one, teach one” to “see one, sim many, do one” [13,14,15]. Classically, the simulations were performed on human and animal corpses, then with the assistance of virtual reality simulators [16]. Along with the development of medical 3D printing, cost-effective, personalized tools and training phantoms began to appear. Torres et al. used CTA to manufacture 3D aortic aneurysm models [17]. Residents were asked to perform endovascular aortic aneurysm repair on 3D replicas, and then in the real environment. The group that trained on patient-specific simulators before going into the operating theatre had better results, including reduced fluoroscopy and surgery time, and lower volume of contrast used. Similarly, the presented GA model has the potential to improve patient safety. All physicians validating the GA model indicated its usefulness in training less-experienced colleagues. 

It is unclear whether there is a need to manufacture physical, patient-specific simulators. The literature shows that standard virtual simulators also improve performance metrics [18]. However, among the many types of intervention available for virtual simulation, there is currently no dedicated training for difficult needle punctures, including the GA puncture. For this reason, the only way to access such tools is to use 3D printing. The advantages of physical, printed simulators include better haptic parameters, such as pushability, torqueability, and trackability [13]. 

According to our best knowledge, this is the first GA model created for clinical purposes. Our model introduces a critical feature—two channels for contrast administration, connected to the hollowed GA. Filling the system with contrast enables the simulation of the endovascular procedure under fluoroscopy. 

By embedding the arteries and bones in transparent silicone, our manufacturing process was very cost-effective. Industrial 3D printers can create such a model in one stage and in a shorter time frame; however, the cost of consumables would be extremely high, mainly because of the massive soft tissue volume, around 15 L. Therefore, a similar model printed with high-grade equipment would cost around $10,000–$12,000. Furthermore, only a few industrial printers are able to produce such models due to the multi-material components and significant building volume. Another significant factor is the price of such equipment, which may exceed $200,000. For comparison, the printer used in our study retails for around $6000. These disadvantages make industrial 3D printers a less viable option. 

Taking into consideration the results of the questionnaire, clinicians found the model useful for training and preoperative planning. These results confirm the feasibility of the proposed model in a clinical environment. It should be emphasized that this data is based on the subjective perceptions of the physicians who tested the model. A clinical study is needed to evaluate hard clinical outcomes. This will most likely have to be performed in multiple high-volume centers to achieve statistical power, as the GA puncture is a relatively rare procedure.

The Likert scale demonstrated the limitations of the proposed model. Some of the physicians considered the needle manipulation unrealistic. It was harder to change the angle of the needle after inserting it into the silicone. The difficulties were due to silicone’s mechanical properties, which are different from those of muscles and subcutaneous tissue. Future models should consider resins, silicones, or biomaterials with physical properties similar to those of soft tissues. Also, fluoroscopic images of bones were not visible. Some considered it to be a shortcoming. In the future, this can be improved by 3D printing bones with a barium-enriched filament [19]. Opaque materials could also be used for further training, such as ultrasound-guided procedures.

## 5. Conclusions

Three-dimensional printing of a cost-effective gluteal artery model for needle puncture simulation is feasible. The phantom can successfully simulate the puncturing of the GA under angiography guidance. It also has the potential to impact resident training and help physicians better understand patient anatomy. Further studies should be conducted to assess the clinical value.

## Figures and Tables

**Figure 1 jcm-09-00686-f001:**
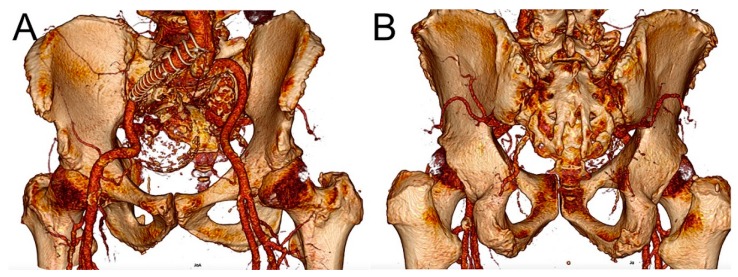
Three-dimensional volume reconstruction of a computed tomography angiography of a patient with type II endoleak to internal iliac artery aneurysm in (**A**) anterior and (**B**) posterior views.

**Figure 2 jcm-09-00686-f002:**
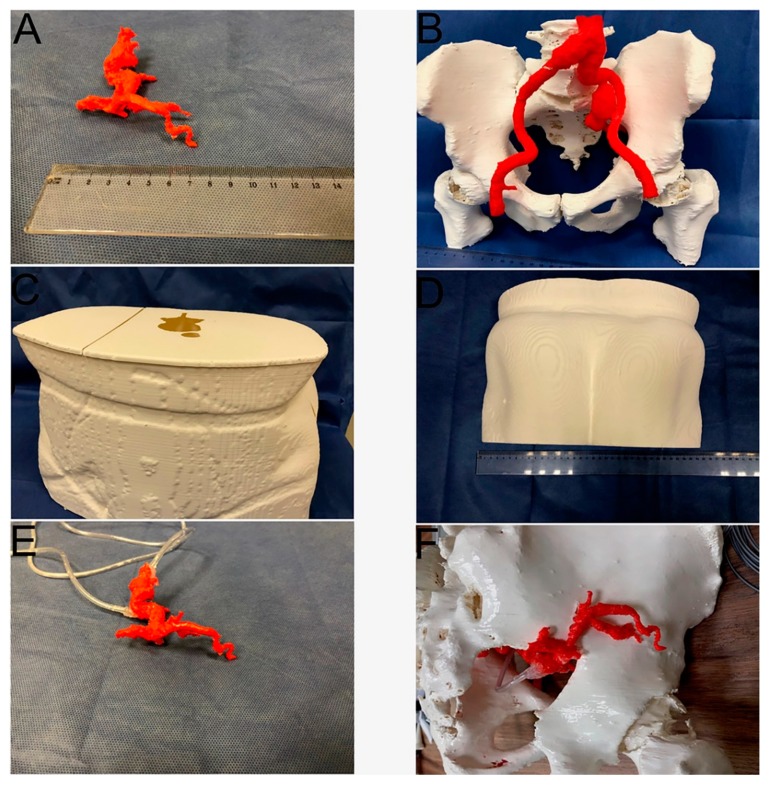
Three-dimensionally printed parts and their assembly. (**A**) Gluteal artery and internal iliac artery aneurysm lumen (**B**) Bones and arterial system (**C**) Skin shell, front and upper part with openings visible; (**D**) Skin shell, rear part (**E**) Gluteal artery model with outlet and inlet ducts (**F**) Gluteal artery model glued to the pelvic bone model.

**Figure 3 jcm-09-00686-f003:**
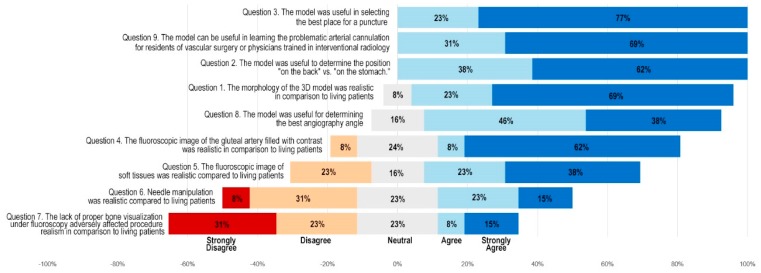
Results of a questionnaire presented as a diverging Likert scale summary.

**Figure 4 jcm-09-00686-f004:**
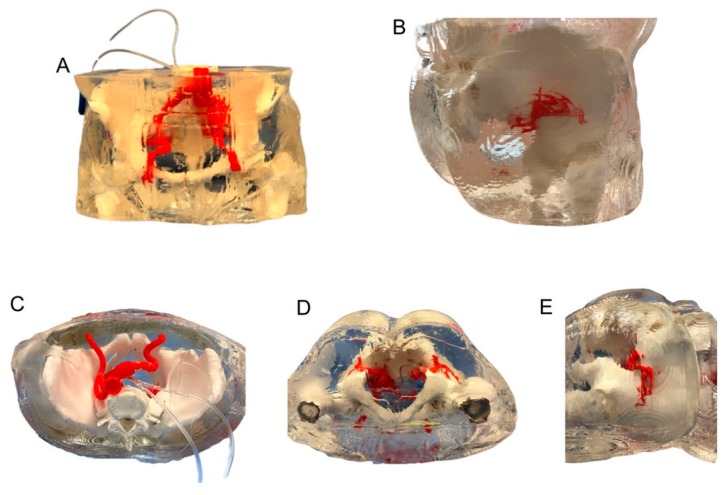
Silicone model. (**A**) Front view (**B)** Lateral rear view (**C**) Top view (**D**) Bottom view (**E**) Side view.

**Figure 5 jcm-09-00686-f005:**
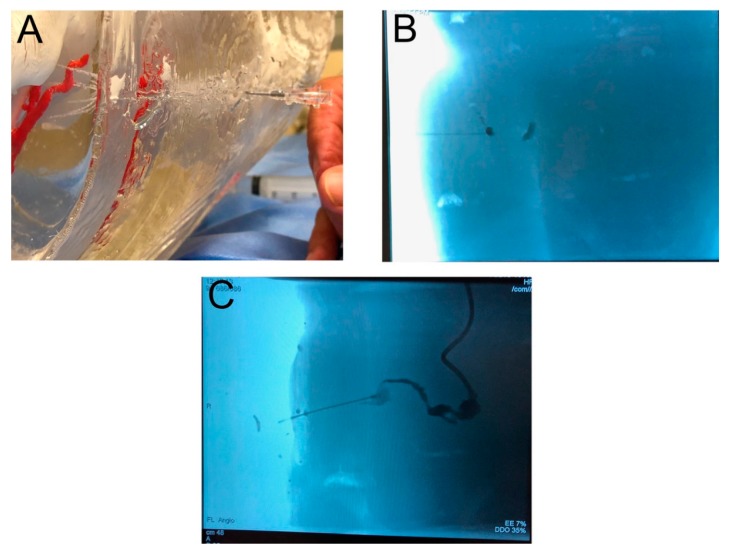
Needle puncture procedure. (**A**) View of puncturing 3D silicone model (**B)** Gluteal artery needle puncture under fluoroscopy (**C**) Gluteal artery filled with contrast.

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
