# Peer review of "Simulation and Training of Needle Puncture Procedure with a Patient-Specific 3D Printed Gluteal Artery Model"

_jcm, 2020, doi:10.3390/jcm9030686_

Round 1
Reviewer 1 Report
The authors should indicate which gluteal artery they are printing. It appears to be the superior gluteal artery.
The standard of English is not acceptable and I encourage the authors to seek assistance from a scientist fluent in this language.
The authors must provide ethical approval or exemption for study and written informed consent of patient. A privacy statement which complies with GDPR would be valuable.
More detail regarding the printing parameters is necessary. Layer thickness, printing temperatures for Filaflex and PLA etc
More detail is required concerning the Shore Hardness of the translucent silicone.
Have the authors tested the model with sonography? An echogenic solution could be injected and visualised.
Author Response
We would like to thank the Reviewer for comments and suggestions. We made major improvements to our manuscript and hope it is now satisfactory.
Comments and Suggestions for Authors
>The authors should indicate which gluteal artery they are printing. It appears to be the superior gluteal artery.
Correct, that’s a good observation. To be precise it is the 3D model of the superior gluteal artery and its division branches. We specified that information in the abstract and Materials and Methods section of our paper.
>The standard of English is not acceptable and I encourage the authors to seek assistance from a scientist fluent in this language.
The manuscript has been proofread by a native speaker.
>The authors must provide ethical approval or exemption for study and written informed consent of patient. A privacy statement which complies with GDPR would be valuable.
We used retrospectively acquired images with no patient identifiers. Thus, in the light of GDPR (especially Recital 26) and Data Protection Working Party’s opinion, we did not acquire patient informed consent.
>More detail regarding the printing parameters is necessary. Layer thickness, printing temperatures for Filaflex and PLA etc
All models were 3D-printed with layer thickness set to 0.2 mm. The printing temperatures varied for these two filaments and were set to 230C and 210C respectively for Filaflex and PLA. Additionally, we heated the bed to 60C. We used moderated extrusion speed for PLA (50mm/s). The general rule for printing flexible filaments is to use slower printing rates in order to avoid material jamming. Therefore, we slowed down to 30mm/s. The second important thing is to decrease the retraction amount (0.5mm in our case). Fan cooling was turned on in both cases.
Initially, we did not report those details in the paper. As the Reviewer definitely knows, printing parameters can vary a lot, depending not only on filament type, but also the manufacturer and 3D printer itself. However, we do see some benefit from providing this information (especially regarding printing flexible materials, which is a nontrivial task). We added the details as above in Materials and Methods section.
>More detail is required concerning the Shore Hardness of the translucent silicone.
Silicone had Shore hardness of 30 A. We added this information to the manuscript in Materials and Methods section.
>Have the authors tested the model with sonography? An echogenic solution could be injected and visualised.
We haven’t tested the model with sonography. We aimed for robust evaluation under fluoroscopy.
Having said that, we agree that it is a good idea to fabricate the 3D model to train US-guided needle punctures. This should be a subject for future studies in the area and we added this suggestion to the Discussion.
-Authors

Reviewer 2 Report
The manuscript by Pawel et al. aims to assess the clinical reliability of a novel 3D, patient-specific silicone gluteal artery model for preprocedural planning and physician training of arterial access. The system was comprised of pelvic bones and the arterial system extrapolated from CT scans into a 3D printable STL file. These components were placed within a patient-specific 3D mold of the pelvic region to create a silicone model for assessment via contrast-assisted fluoroscopy puncture. The developed model was assessed by thirteen endovascular specialists, majority of which had over 10 years of experience, using a survey followed by Linkert Scale analysis. Overall, the developed model was shown to be useful as a preoperative and training tool to determine optimal puncture site and patient position during the procedure.
One issue with the methods proposed in this work was the allowance for some clinicians to view the silicone model prior to puncture testing. Because the silicone is clear the underlying vascular anatomy was visible and therefore does not truly assess the clinician’s ability to puncture the gluteal artery in a clinically relevant manner. An alternative method such as using an opaque silicone or other material for the skin/overlying tissue would be better suited for test the model’s validity.
While I do believe the patient-specific model developed in this work is novel, the low sample size and methods used do not provide enough evidence to show the products clinical relevance. Furthermore, the main advantage of the proposed model was for preoperative planning and this was not tested. Additional work should be required assessing the use of patient-specific gluteal artery models for preop planning and the effect it has on performing the puncture in clinic.
Author Response
We would like to thank Reviewer for the comments. Please find our point-by-point response below.
Comments and Suggestions for Authors
The manuscript by Pawel et al. aims to assess the clinical reliability of a novel 3D, patient-specific silicone gluteal artery model for preprocedural planning and physician training of arterial access. The system was comprised of pelvic bones and the arterial system extrapolated from CT scans into a 3D printable STL file. These components were placed within a patient-specific 3D mold of the pelvic region to create a silicone model for assessment via contrast-assisted fluoroscopy puncture. The developed model was assessed by thirteen endovascular specialists, majority of which had over 10 years of experience, using a survey followed by Linkert Scale analysis. Overall, the developed model was shown to be useful as a preoperative and training tool to determine optimal puncture site and patient position during the procedure.
To address the Reviewer’s concerns that the conclusions are not supported by the results, we have changed the wording of our Conclusion section and rephrased the goal of our study. We also improved other sections of the manuscript, including description of methods.
One issue with the methods proposed in this work was the allowance for some clinicians to view the silicone model prior to puncture testing. Because the silicone is clear the underlying vascular anatomy was visible and therefore does not truly assess the clinician’s ability to puncture the gluteal artery in a clinically relevant manner. An alternative method such as using an opaque silicone or other material for the skin/overlying tissue would be better suited for test the model’s validity.
Thank you for your comments and valuable insights. While we do agree that using non-transparent silicone might be beneficial in some scenarios, our idea for the study was slightly different. We specifically looked for transparent silicone to allow trainees visualization of 3D structure. It was important for us to help them understand three-dimensional location of gluteal artery and its spatial relationships. As this model was patient-specific, our study aimed to show a proof of concept where person performing the procedure can learn patient’s anatomy. We would absolutely consider using different material, as suggested by Reviewer, if this was more general training phantom. We appreciate your feedback and agree that this should be explored in the future, preferably in a different study scenario.
We added a short information about potential benefits of using opaque materials in the Discussion section.
While I do believe the patient-specific model developed in this work is novel, the low sample size and methods used do not provide enough evidence to show the products clinical relevance. Furthermore, the main advantage of the proposed model was for preoperative planning and this was not tested. Additional work should be required assessing the use of patient-specific gluteal artery models for preop planning and the effect it has on performing the puncture in clinic.
Thank you for this comment. In the revised Discussion of our manuscript, we explain the limitations of our study further, including lack of hard clinical outcomes.
We would like to point out that although models fabricated with proposed method are dedicated for preoperative planning, main strength of the paper is the methodology itself. Similar models have not been presented in the literature, especially considering our low-cost approach. Our study’s objective was to make sure that the models are actually feasible to use clinically. Evaluation of specific clinical endpoints should be -- as you mentioned -- performed on a large sample and preferably in a multi-institutional study. Here, we completed a number of novel and not described previously experiments: from developing a full methodology of 3D printing the model, administering the contrast into the model, evaluating the visibility under angiography and extensive analysis of model realism with clinicians. We hope this is a solid foundation for clinical studies in the future. It’s also important to remember that GA puncture is a rare procedure. This underlines the educational value of presented models.
- Authors

Round 2
Reviewer 1 Report
The authors have responded satisfactorily to all points raised.
Reviewer 2 Report
Significant improvements were made in the revision that clarified the novelty and direction of this work and warrants acceptance. Specifically, additions made in the discussion section and discussed in the authors comments resolved original concerns about the evaluation of the product on clinical outcomes. Additionally, more emphasis on the methods and application as patient-specific visual model were introduced throughout the article that focuses the reader’s attention on the authors main purpose of the study.